# Soil carbon sequestration accelerated by restoration of grassland biodiversity

Yi Yang[1], David Tilman [1,2], George Furey[1] & Clarence Lehman[1]

Agriculturally degraded and abandoned lands can remove atmospheric $CO_2$ and sequester it as soil organic matter during natural succession. However, this process may be slow, requiring a century or longer to re-attain pre-agricultural soil carbon levels. Here, we find that restoration of late-successional grassland plant diversity leads to accelerating annual carbon storage rates that, by the second period (years 13–22), are 200% greater in our highest diversity treatment than during succession at this site, and 70% greater than in monocultures. The higher soil carbon storage rates of the second period (years 13–22) are associated with the greater aboveground production and root biomass of this period, and with the presence of multiple species, especially C4 grasses and legumes. Our results suggest that restoration of high plant diversity may greatly increase carbon capture and storage rates on degraded and abandoned agricultural lands.

[1] Department of Ecology, Evolution, and Behavior, University of Minnesota, St. Paul, MN 55108, USA. [2] Bren School of Environmental Science and Management, University of California, Santa Barbara, CA 93106, USA. Correspondence and requests for materials should be addressed to D.T. (email: tilman@umn.edu)

Soils store climatically significant amounts of carbon (C) as soil organic matter, globally about 2.3 times greater than the C in atmospheric $CO_2$ and 3.5 times greater than the C in all living terrestrial plants[1]. However, prolonged cultivation accelerates the decomposition of soil organic matter and can cause the loss of 20–67% of the soil C in an agricultural field[2–4]. Between 1850 and 1998, global agricultural cultivation led to the release of ~78 Gt of C from soil as $CO_2$ to the atmosphere[4], with ~133 Gt of soil C so released since the beginning of agriculture[5]. Since the current global annual $CO_2$ emissions from fossil fuels and all other sources are ~10 Gt of C[6], soil C sequestration has thus been proposed as a plausible partial climate mitigation strategy that might buy time while low-carbon technologies are being developed and adopted[7]. Indeed, a recent international initiative has set a target of increasing global soil organic matter by 0.4% per year to help negate some greenhouse gas emissions[8].

Abandoned agricultural lands have been a particular area of interest for carbon capture and storage[9–12] because of their high-potential capacity for C sequestration[8]. An estimated ~430 million hectares of land globally has been cleared, cropped, degraded and then abandoned[13]. When agriculturally degraded lands are abandoned and undergo ecological succession, they remove atmospheric $CO_2$ and sequester its C as soil organic matter[7,14]. However, this process may require a century or more for soil carbon to re-attain pre-agricultural levels[15–17]. Since the original soils of these abandoned lands had been formed by native, late-successional and often highly diverse ecosystems, we decided to test the possibility that rapid restoration of late-successional plant diversity might accelerate soil C storage above the rates observed during natural succession[18]. Here, we report how the experimental restoration of different levels of late-successional plant diversity on abandoned agricultural land impacted the rate of soil C storage across 22 years, and how these rates compare to those observed during succession at the same site.

During the ecological succession that follows abandonment of agricultural lands at our research site in Minnesota, USA, ~50 years are required for the perennial plant species that dominate nearby native grassland ecosystems to become dominant[19,20]. During the first decades of succession, abandoned agricultural lands are dominated by annual plants and fast-growing and fast-dispersing C3 plant species[20]. These species are gradually out-competed by late-successional perennial prairie C4 grass species that are strong competitors for soil nitrogen because of high-root mass, but that are slow to arrive in a field, and slow to spread across it, because they have low-dispersal rates and low-growth rates[21–23]. Because the C3 grass and forb species have both less roots and roots that, in a 1 year period, decomposed 55% and

138% faster, respectively, than did roots of the C4 grasses[24], it seems plausible that soil C storage rates would increase once native C4 grasses attain dominance during succession. In contrast, our experiment, started in 1994 to understand effects of plant biodiversity, had effectively restored various levels of late-successional plant diversity in replicated plots planted on a highly degraded soil within 5 years. The plant diversity restoration treatments represent planting either 1, 2, 4, 8, or 16 species of perennial grassland plants common in nearby undisturbed native prairie, with ~30 replicate plots for each of these 5 levels of planted diversity[25]. All plots were sampled periodically for soil C, root C and species abundances, with the upper layer of soil (0–20 cm for soil C and 0–30 cm for roots) sampled more frequently than deeper layers (20–60 cm for soil C and 30–60 cm for roots) (see Methods). We report both soil C and root mass because numerous analyses have shown that grassland plots with higher root mass tend to accumulate soil C at greater rates[24,26].

In our 22-year experiment, annual rates of soil carbon storage increased through time and, on average across all diversity treatments, were ~90% greater in the second period (13–22 years) than in the first period (1–13 years). The highest diversity treatment had carbon storage rates in the second period (13–22 years) ~200% greater than during succession at this site. Across the full time span, the highest diversity treatment stored 178% more C in soil than did the monocultures, demonstrating the potentially large carbon storage advantage that rapid restoration of high plant diversity may provide. These higher rates of soil carbon storage were strongly associated with the joint presence of C4 grasses and legumes in higher-diversity plots. Such plots also had greater aboveground production and root biomass. In total, our results suggest that both high plant diversity and the presence of specific combinations of plant functional traits may be needed to maximize the rate of below-ground carbon storage on degraded and abandoned agricultural lands.

## Results

**Soil carbon storage rates and plant diversity.** We found that, at each experimentally imposed level of plant diversity, the average annual rate of C storage in soils, as quantified by $\Delta C/\Delta t$ (units of Mg of C $ha^{-1}$ $y^{-1}$), was greater in the second period (13–22 years) of the experiment than in the first period (1–13 years; Fig. 1a, b). These accelerating rates of soil C sequestration were apparent for both the 0–20 cm depth soil profile (Fig. 1a) and the full 0–60 cm profile (Fig. 1b). On average over all diversities, annual storage rates for the second period were 88% and 253% greater than for the first period for the 0–60 cm and the 0–20 cm profiles, respectively. In addition, across the 5 times that the top 20 cm of soil was sampled

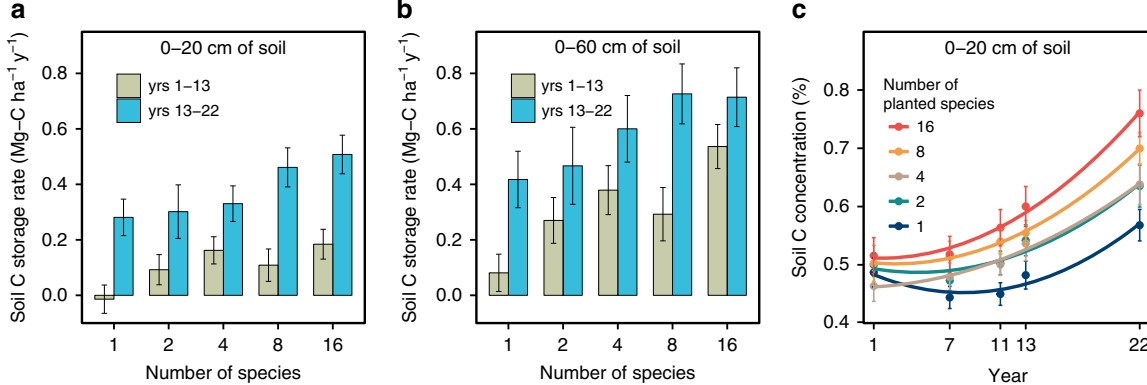

**Fig. 1** Change in soil C over 22 years. **a, b** Average annual soil C storage rates over years 1–13 (green bars) and years 13–22 (blue bars) in upper 20 cm of soil (**a**) and in upper 60 cm (**b**) (Supplementary Table 1). Bars are means with standard errors. **c** Dynamics of soil C concentration in upper 20 cm of soil for plots planted with 1, 2, 4, 8, or 16 perennial grassland species (Supplementary Table 2). Dots are means with standard errors; fitted curves are quadratic

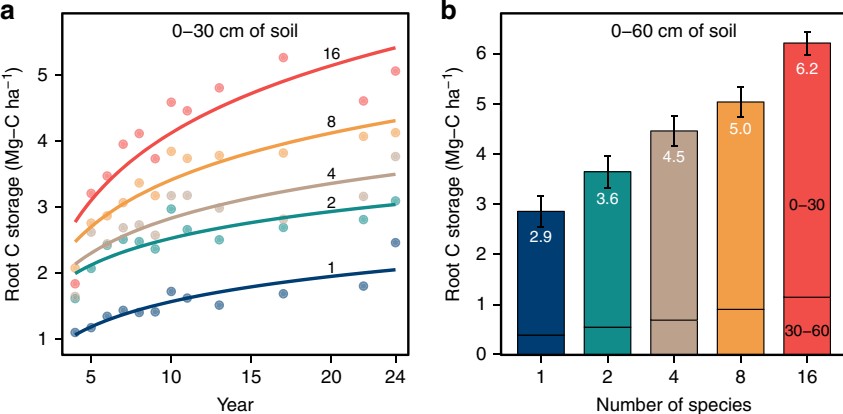

**Fig. 2** Change in root C over 24 years. **a** Change in root C in upper 30 cm of soil under different experimentally imposed levels of plant species diversity. Dots indicate mean root C at a given year; curves fitted with log functions; the number on each curve indicates plant species diversity. **b** Total root C storage after 24 years of growth in upper 60 cm of soil. Numbers in white indicate mean total root C storage, error bars indicate standard errors, and numbers in black indicate soil depth increments (cm)

for C, the time dynamics of soil C concentration at each level of plant diversity was best fit by quadratic equations curving upward (Fig. 1c), further demonstrating that rates of C sequestration accelerated through time.

Rates of soil C sequestration were greater at higher plant diversity (Fig. 1). Annual storage rates for the first period (1–13 years) for the 0–60 cm soil depth profile were 0.08 (±0.07), 0.27 (±0.08), 0.38 (±0.09), 0.29 (±0.10), to 0.54 (±0.08) Mg-C ha$^{-1}$ y$^{-1}$ in the 1-, 2-, 4-, 8-, and 16-species treatments, respectively (Fig. 1b; Supplementary Table 1). For the second period (13–22 years), they increased to 0.42 (±0.10), 0.47 (±0.14), 0.60 (±0.12), 0.73 (±0.11), and 0.71 (±0.11) Mg-C ha$^{-1}$ y$^{-1}$, respectively (Fig. 1b; Supplementary Table 1). For the full 22-year duration of the experiment, when compared to means across all species in monocultures, higher plant biodiversity led to 60%, 115%, 115%, and 178% greater soil C storage in the 2- to 16-species plots, respectively, for the 0–60 cm profile (Supplementary Table 3). When initial soil C levels are considered, annual soil C proportional growth rates (dC/d$t$ * 1/C) were 0.6%, 1.0%, 1.3%, 1.3%, and 1.6% y$^{-1}$ for the 1, 2, 4, and 16 species treatments, respectively, across the full experimental duration for the 0–60 cm profile (Supplementary Table 4).

A linear mixed model showed that soil C concentration (log-transformed) for the 0–20 cm soil depth increased through time with the quadratic time term being positive and significant ($P <$ 0.0001), and also increased with the number of planted species ($P = 0.0025$), and had a positive time × diversity statistical interaction ($P < 0.0001$; Fig. 1c and Supplementary Table 5). When comparing the annual soil C storage rates of the first period (1–13 years) to those of the second period (13–22 years) for the 0–60 cm soil depth (Fig. 1b), multiple regression (overall: $F_{3,300} = 10.78$, $P < 0.0001$, $r^2 = 0.10$; Supplementary Table 6) showed that storage rates (Mg-C ha$^{-1}$ y$^{-1}$) were higher in the second period ($P < 0.0001$), were positively associated with diversity (the number of planted species; $P = 0.0002$), but the period × diversity interaction was not significant ($P = 0.6939$; Fig. 1b). This same pattern held for soil C storage rates for these 2 periods for the 0–20 cm depth (overall: $F_{3,300} = 19.26$, $P < 0.0001$, $r^2 = 0.16$; Fig. 1a; Supplementary Table 7).

**Aboveground and below-ground productivity and diversity.** Root C and the amount of aboveground plant biomass produced each year (productivity) increased with plant diversity. After increasing, especially at higher plant diversity, for the first 8 years, aboveground productivity subsequently had year-to-year variation that corresponded with growing season precipitation and

temperature conditions (Supplementary Fig. 1). In contrast, root mass (0–60 cm depth), which was on average 5.5 times the mass of aboveground biomass, tended to increase throughout the 24 years, but at decelerating rates (Fig. 2a). By the 24th year of the experiment, plant roots in the 0–60 cm depth contained an average of 2.9 (±0.3) Mg-C ha$^{-1}$ in monoculture plots and 6.2 (±0.2) Mg-C ha$^{-1}$ in 16-species plots (Fig. 2b). Most root C was concentrated in the upper 30 cm of soil, with 16–23% more root C between 30 and 60 cm for low- to high-diversity plots (Fig. 2b). A linear mixed model showed that root C (0–30 cm) increased with time, with the number of planted species, and with their positive interaction ($P < 0.0001$ for all estimates; Supplementary Table 8). A linear mixed model showed that the root:shoot ratio (R:S; root biomass (0–30 cm depth)/aboveground biomass) of plots was statistically independent of plant diversity ($P = 0.0701$), increased through time ($P < 0.0001$), was significantly higher when C4 grasses were present (least square mean; R:S = 5.0) versus absent (R:S = 2.5; $P < 0.0001$) and when C3 grasses were present (R:S = 4.0) versus absent (R:S = 3.6; $P = 0.0293$), but was lower when legumes were present (R:S = 2.8) versus absent (R:S = 4.8; $P < 0.0001$) and when nonlegume forbs were present (R:S = 3.4) versus absent (R:S = 4.2; $P = 0.0002$; Supplementary Table 9). For the last 5 years of the experiment, the average R:S ratio (0–30 cm soil depth) across all plots was 4.3, and was 5.5 for 0–60 cm depth roots for the 3 years root biomass was sampled to the depth of 60 cm (in 2006, 2015, and 2017).

**Comparison with natural succession in nearby old fields.** During succession in abandoned agricultural fields at our site, annual and short-lived perennial plant species were dominant initially[19]. The native perennial plant species that dominate undisturbed grasslands at our site are rarely present during the first decade of succession, and colonize into and increase in abundance slowly through time, with native species and native C4 grasses comprising about 90% and 55%, respectively, of the plant community abundance after ~50–70 years of succession[19,20]. Analysis of soil C along this chronosequence (based on ~2000 plots from 21 agricultural fields abandoned at different times from 4 to 74 years ago), gives an annual rate of C storage of 0.17 (±0.05) Mg-C ha$^{-1}$ y$^{-1}$ for the 0–20 cm depth profile and suggests that this annual rate of soil C accumulation under natural succession is approximately constant across this 70 year period[20]. During the first period (1–13 years) of our experimental restoration, C storage rates were 0.16 (±0.05), 0.11 (±0.06), and 0.18 (±0.05) Mg-C ha$^{-1}$ y$^{-1}$ in the 4, 8, and 16 species treatments for the 0–20 cm depth

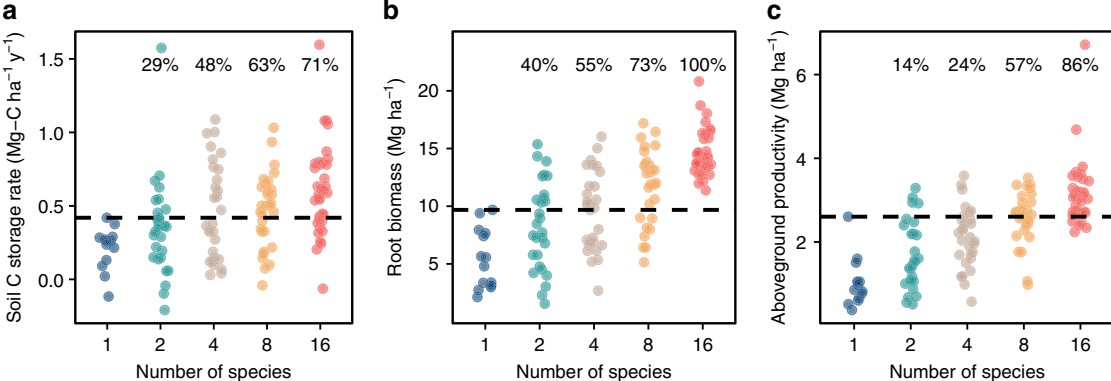

**Fig. 3** Higher-diversity plots versus the best species in monoculture. **a** Soil C storage rate over the entire 22 years for the 0–60 cm soil profile. **b** Mean root biomass for the 0–60 cm soil profile (average of 2006, 2015, and 2017). **c** Mean productivity from 2012 to 2016. The numbers at the top of each panel are proportions of mixture plots surpassing the best performing monoculture species. Proportions increase with diversity for all three measures. Monocultures values are means of plots planted to a given species

profile (Supplementary Table 1), comparable to those of succession. However, during the second period (13–22 years), we observed an acceleration of soil C storage, with rates of 0.33 (±0.06), 0.46 (±0.07), and 0.51 (±0.07) Mg-C ha$^{-1}$ y$^{-1}$ in the 4, 8, and 16 species treatments for the 0–20 cm depths (Supplementary Table 1). These rates are 94%, 170%, and 200% greater, respectively, than those observed during natural succession at our site. The magnitude of this effect suggests that active restoration of abandoned croplands to high diversity of late-successional plant species might triple the annual rate of soil C storage, and thus provide a potentially important climate moderating effect.

**Biodiversity and sampling versus complementarity.** Increases in biodiversity can impact ecosystem functioning because of sampling effects (the greater likelihood of the functionally best species being present at high diversity), or because of complementary interactions among species caused by their differing traits[27]. The signature of sampling effects is that high-diversity plots never perform significantly better than the single best species in monoculture. In contrast, the signature of complementarity is that, at higher diversity, an increasingly larger number of plots have functioning that exceeds that of the best species in monoculture[25,28]. Comparing higher diversity plots with the performance of the monocultures of the best species offers strong evidence for a complementarity effect, rather than a sampling effect (Fig. 3). For soil C storage rates, root biomass, and aboveground biomass yield, the proportion of 2-, 4-, 8-, 16-species plots surpassing the best performing monoculture species (mean of plots with the same species) increases with diversity. In particular, the majority of 8- and 16-species plots outperformed the best monocultures for all 3 variables. For root biomass (Fig. 3b), all of the 16-species plots have higher biomass than the best monoculture species, demonstrating that no species on its own produces as much below-ground biomass as do the highest diversity plots. Although this analysis does not prove the underlying causative mechanisms, it demonstrates that soil carbon stores, root mass, and aboveground productivity are greater at higher diversity because of some form of interspecific interaction or complementarity, rather than a simple sampling effect.

**Mechanisms explaining soil C storage.** Soil C sequestration was positively associated with aboveground plant biomass and root biomass. A linear mixed model showed that soil C concentrations (as %) in the upper 20 cm of soil were positively correlated with root biomass ($P = 0.0004$), time ($P < 0.0001$), and aboveground plant biomass ($P = 0.0033$; Supplementary Table 10). We further

investigated which species may have contributed to soil C sequestration using species abundance data we have collected periodically since 2001. A multiple regression analysis of the soil C storage rate for the 0–60 cm soil profile from 1994 to 2016 (Mg-C ha$^{-1}$ y$^{-1}$) as a function of mean species abundances over the past 10 years (2006–2015) showed soil C storage rates were significantly correlated with seven species ($P < 0.05$, Supplementary Table 11), including three C4 grass species, two legume species, a C3 grass species and a forb species. These results suggest that a diversity of plant functional traits were involved in soil C storage[24]. Moreover, at each level of plant diversity, plots containing both one or more C4 grass species and one or more legume species had greater root biomass and soil carbon storage than did plots with C4 grasses but no legumes, and those with legumes but no C4 grasses (Fig. 4a, b)[24]. This may be because, of the four functional groups, C4 grasses have the greatest root mass (Fig. 4c), produce roots with the lowest decomposition rates (Fig. 4d), and are the dominant functional group (45% of aboveground biomass). In contrast, legumes fix N but have lower root mass, and produce litter that decomposes more rapidly. When these differing traits are combined in a plot, we suggest that rates of soil C storage are elevated because C4 grasses can use N released by decomposing legume roots to produce above- and below-ground biomass that decomposes slowly, causing C to accumulate. Higher-diversity plots that contain more species of C4 grasses and legumes have more above- and below-ground biomass which likely contributes to their greater rates of C storage in soil.

## Discussion

These results show that, during the second period (13–22 years) of our experiment, the active restoration of high-diversity late-successional plant communities caused soil C accrual to accelerate, with C pools accumulating at a rate 2–3 times that observed in natural succession at our site. We suggest that these elevated rates resulted from how rapidly we re-established, relative to succession, high-diversity communities populated with perennial plant species with high root:shoot ratios and low decomposition rates of these roots.

The increases in soil C sequestration rates that we observed following restoration of highly diverse plant communities suggest that similar studies should be performed in other sites to determine if or how differences in climate, soil type, plant functional traits may influence soil C storage rates on abandoned lands. It is important to note that the soils of our site have >90% sand, low-organic matter (~0.5%), and limited horizon development, and

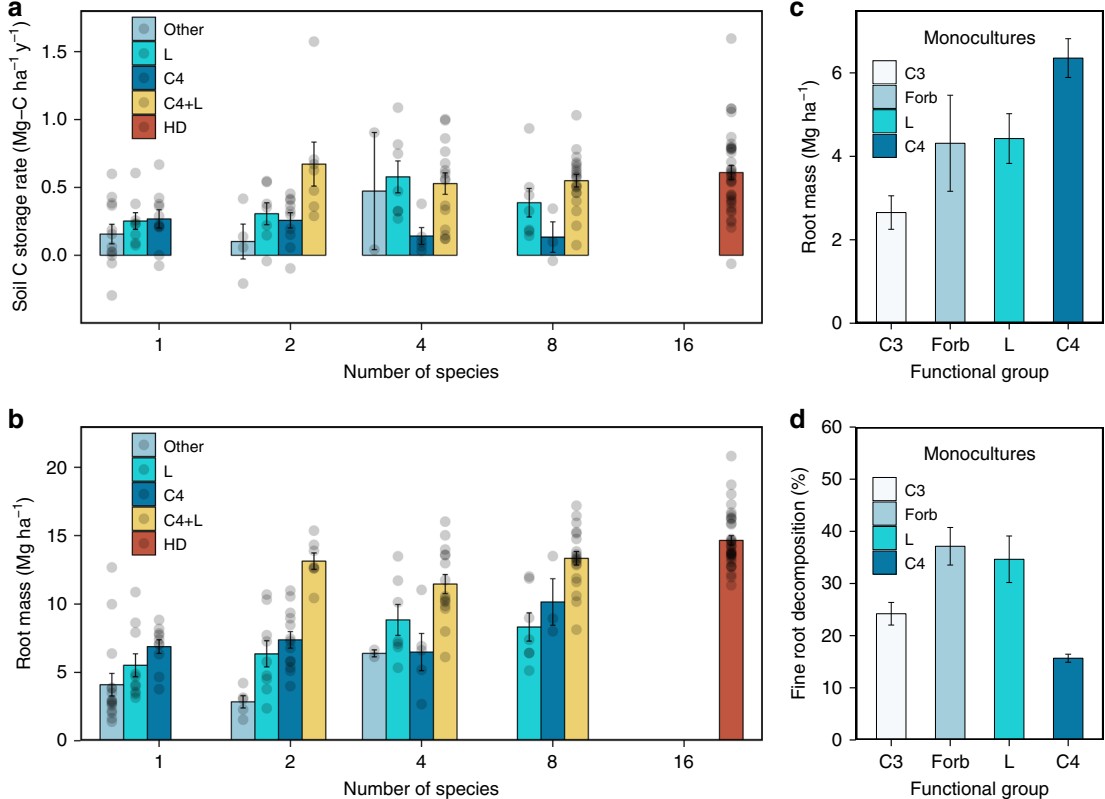

**Fig. 4** Functional composition and traits influence root biomass and soil C storage. **a** Soil C storage rates over the entire 22 years for the 0–60 cm profile. **b** Mean root biomass for the 0–60 cm soil profile (average of 2006, 2015, and 2017). **c** Mean root biomass of different functional groups in monoculture plots (0–30 cm soil profile, average of 2006, 2015, and 2017). **d** Fine root decomposition percentage of different functional groups (measured after 10 months of field incubation, which included ~5 winter months[54]). In all panels, bars are means with standard errors. In **a** and **b**, dots indicate plot results: C4—plots with at least one C4 but without legume; L—plots with at least one legume but without C4; C4 + L—plots with at least one C4 and one legume; Other—forbs, C3, or woody; HD—16-species plots, which include both C4 and legume (of typically 3–4 species each). In **c** and **d** for monoculture plots, L means legume

were subject to herbicide, some soil removal, plowing and disking prior to planting. Moreover, our plots were weeded several times every year to retain their desired diversity and composition. As in other experiments[29,30], weeds appeared much more frequently and in higher density in our lower-diversity plots. Weeding removes biomass, and thus might lower soil C storage rates. However, if competition with weeds reduces the productivity of the planted late-successional species[31], weeding might increase soil C storage because of greater abundance of late-successional species. In a parallel experiment at our site, plots that had been planted with 32 late-successional grassland species and never weeded had remarkably similar rates of soil C sequestration as our 16-species plots (Supplementary Fig. 2), suggesting that high diversity of late-successional species leads to high rates of soil C storage even without weeding. Finally, our experiment used native species, while nonnative and highly productive C4 grasses, such as *Miscanthus*, may also lead to high soil C sequestration rates[32]. The use of non-native species, however, may pose risks of invasions that need to be better understood and considered[33,34].

When compared with natural succession at our site, the high rates of soil carbon accumulation observed in our restored high-diversity grassland ecosystems suggest that immediately restoring degraded land to high diversity of the dominant late-successional perennials may notably increase the ability of these lands to contribute to climate change mitigation via C sequestration[35]. High-diversity restorations on abandoned and degraded lands may provide other environmental benefits, including reduced nitrate leaching[36], reduced year-to-year variability in biomass

harvests[37], fewer invasive plant species[38], and lower soil N2O emissions[39,40]. Restorations also provide habitat that might help lower extinction risks[41]. Another climate moderation benefit could occur if aboveground biomass from degraded and abandoned agricultural lands were harvested for bioenergy production[42–44]. In total, the restoration of high-diversity ecosystems on degraded and abandoned lands merits further consideration for its potential to provide multiple ecological and environmental benefits, including increased carbon capture and sequestration.

## Methods

**Experimental design**. Our experiment was located in an abandoned agricultural field at Cedar Creek Ecosystem Science Reserve, Minnesota, USA[25]. In 1993, the field was treated with the herbicide glyphosate and then burned once its herbaceous vegetation had died, had the top 6–8 cm of soil removed to reduce seed bank, and was plowed and repeatedly harrowed. In 1994, 168 plots were established, each 13 m × 13 m (later reduced to 9 m × 9 m). They were randomly selected for restoration to 1 of 5 different diversity treatments: planted with 1, 2, 4, 8, or 16 perennial grassland/savanna species. Species in each plot were randomly selected from a set of 16 grassland species composed of 4 species in each of 4 different functional groups: C4 grass species, C3 grass species, legume species, and nonlegume herbaceous forb species. In addition, two different species of savanna oaks were included in this set, but annual spring burns caused these oaks to be extremely rare, and then lost from the experiment (Supplementary Table 12). All plots received 10 g m$^{-2}$ of seed in May 1994 and 5 g m$^{-2}$ in May 1995, with seed mass divided equally among the species planted in a given plot. Plots were burned annually in spring before growth began to imitate natural fires that were common in these grasslands. Plots have never been fertilized and have been weeded three or four times a year to maintain the intended species compositions and diversity. Current species richness and diversity, as indicated by the effective number of species[45,46], are highly corrected with the number of plant diversity ($r^2 = 0.64$;

Supplementary Fig. 3). Additional details on the experiment design have been published previously[24,25,47].

Soil C samples from the upper 0–20 cm soil depth and 2.5 cm in diameter, were collected, in each of 9 sites per plot, 5 times throughout the project—in 1994 before planting, 2000, 2004, 2006, and 2015. Additional soil C samples to 60 cm of depth were collected in 1994, 2006, and 2015. The samples for each plot were first sieved to remove roots; the nine samples per plot were then combined, by depth, mixed, dried, mixed again, and then subsampled for grinding and archiving. Before analysis for total C, they were dried again for 5 days in glass vials, and analyzed for total C by combustion and gas chromatography (Costech ECS 4010 Analyzer, Costech Analytical Technologies Inc., Valencia, CA). Soil dry weights and bulk density were determined after drying soils at 105 ºC.

Below-ground biomass between 0 and 30 cm was periodically sampled over the past 24 years (Fig. 2a) via collection of 12 soil cores, 5 cm in diameter, evenly spaced across each plot. Deeper soil samples, 30–60 cm deep, were similarly collected in 2006, 2015, and 2017. Roots in the cores were separated from soil by rinsing on a fine mesh screen with a gentle water shower until the soil largely had been removed. The bare roots were then dried at 40 ºC, placed in a sieve to retain roots and dislodge any remaining soil. The clean dried roots were weighed. Root C was calculated as 40% of the dry biomass, based on analyses of a subset using the Costech ECS 4010 Analyzer.

**Bulk density measurement**. We measured bulk density in June, 2017, in 87 plots that had been chosen in a stratified-random sampling design for each level of the species diversity treatment. In particular, we randomly chose 16 plots planted with 2 species, and 15, 15, and 15 plots planted with 4, 8, and 16 species richness, respectively. We chose 26 monoculture plots randomly but subject to the constraint that 2 plots were chosen for each of the 12 herbaceous grassland species that have persisted at higher abundances since the initial planting. To measure bulk density, one core per plot was taken to a depth of 60 cm using an AMS Inc. (American Falls, ID, USA) split soil core sampler with a removal jack (part numbers 400.99, 403.41, 403.73, 211.05, and 211.06). To prevent sand from making it difficult to separate the parts of the corer, polytetrafluoroethylene thread seal tape was used to wrap all threads each time, and then was carefully removed. Each core was taken >1 m from the outside of the plot to avoid an edge effect. Upon removal, the sampler was split open and the soil core was cut into three sections: 0–20, 20–40, and 40–60 cm. These soil sections were then placed in resealable plastic bags before drying. In some cases, a small amount of soil was lost from the bottom of the core and the length of the remaining core was used for volume calculations. Most cores remained intact due to recent rains, negating the need to use the excavation method. When soil was lost from the bottom of a core, we assumed, based on a previous soil survey[48], that bulk density was constant and independent of depth for depths greater than 40 cm. For a subset of plots, the total length of the whole core was compared with the depth of the hole to estimate compaction. There was a mean compaction of 4.5%. An ordinary least squares regression was conducted with compaction as the $y$-variable and species richness as the $x$ variable and there was no relationship ($F_{1,81} = 0.004$, $P = 0.948$). We also had tested the excavation method (a plastic sheet was used to line a hole after a core at each depth was taken and then filled with a known volume of water), and decided not use it because it had greater variance. After sample collection, the complete sample from each depth was dried in aluminum bread tins in an industrial drying oven at 105 ºC and then weighed. The final dry weight was used to calculate bulk density at each sampling depth using the appropriate volume. For final calculations, a diameter of 4.8 cm representing the inside sleeve of the corer was used for the final volume calculation. Supplementary Table 13 summarizes mean bulk density results for different species combinations at different depths.

**Estimates of soil C stocks**. We used the measured mean bulk density for each diversity treatment (Supplementary Table 13), together with soil C concentration, to estimate soil C stocks in 1994, 2006, and 2015. For the 20–60 cm of soil, we applied the standard fixed depth approach assuming a constant bulk density for each species diversity level. For the 0–20 cm of soil, we applied the equivalent soil mass approach[49–51] to account for bulk density changes in this layer of soil given the substantial increase in soil C concentration[52,53], especially for the high-diversity plots (Supplementary Tables 2 and 13; Supplementary Fig. 4). Earlier measurements of bulk density in this field and experiment[20,24,48], and the analyses of Supplementary Fig. 4, give a bulk density of the 0–20 m cm depth soil in this field at the time of planting of 1.5 g cm$^{-3}$. Next, we estimated the bulk density in 2006 based on the observed linear decrease in bulk density, as calculated in Supplementary Fig. 4a. Last, we estimated the additional depth needed in 2006 and 2015 to make the soil mass in each year equivalent to that in 1994 for the top 20 cm soil, assuming the soil bulk density and soil C concentration for the additional depth equal the mean of those for the 0–20 cm and 20–40 cm soil profiles. Estimates of soil C stocks in 2006 and 2015 are summarized in Supplementary Table 3.

**Statistical analysis**. In this study, we used the linear mixed model (Restricted Maximum Likelihood, JMP Pro 13.1) for variables involving multiple measurements per plot over time, with plot as a random effect and year as a variable. For these models we report the fixed effect tests for different variables. We used multiple regression in other analyses. All figures were made using R (https://www.r-project.org/). Statistical information is provided throughout the text and in figure captions and Supplementary Tables.

## Data availability

All data used in our analyses can be found at the Cedar Creek Ecosystem Science Reserve website, http://www.cedarcreek.umn.edu/research/data. Data on the multi-diversity restoration are part of the "e120" experiment of the Cedar Creek Long-Term Ecological Research program; data on natural succession are from the "e014" and "e054" experiments. Data on soil C concentration for the 32-species plots are from the "e248" experiment.

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

## Acknowledgments

We thank the Global Climate and Energy Project (GCEP) and the NSF LTER program (DEB-0620652 and DEB-1234162) for funding this research, Troy Mielke for coordinating data collection, and Dan Bahauddin for data management.

## Author contributions

Y.Y. led the data analysis and writing efforts. D.T. established the experiment, contributed to data analysis and writing, and obtained NSF funding. G.F. measured soil bulk density and contributed to data analysis and writing. C.L. obtained GCEP funding and contributed to writing.

## Additional information

**Competing interests:** The authors declare no competing interests.

