## [Peer Review File · Nature Communications]

Reviewers' comments:

Reviewer #1 (Remarks to the Author):

This manuscript describes soil C stocks over 22 to 24 years in an old field in MN, USA. The site is part of a well-known research project where researchers attempted to manipulate plant species richness to determine the effects of richness on a suite of ecosystem characteristics. The species richness treatments are repurposed here as a "restoration" experiment. In the present study, soils were sampled four times post treatment and root biomass was sampled three times. The authors argue that the more species-rich plots had more soil C, and thus that enhanced biodiversity during native species restoration leads to greater soil C sequestration.

The manuscript left me with several questions. First, the authors report C stocks, but not bulk density or C concentrations. Neither are mentioned in the manuscript. Soil C stocks can vary as a result of C concentrations, bulk density, or both. Thus, the absence of these data is a critical omission. Bulk density and C concentrations could vary across space, time, and treatments. Rates and not stocks are reported in figure 1, but how the rates were determined is not reported. I could not determine how many samples were collected for each depth at each time point. Soils were dried that 40 oC; were dry weights determined at the standard 105 oC? If not, masses could differ as a result of moisture retention in more organic or clay-rich sites, skewing the results.

The authors report species richness of treatments that were established in 1994. However, no information is given about whether species richness and composition treatments were maintained throughout the whole experiment. No information is given about the density of individuals, or the functional groups or species traits within and across treatments. The authors do not explore the potential for sampling effect (the potential to select a species with traits that heavily influence the results in higher richness treatments). This could easily influence the results for soil C and root biomass. The authors also do not consider the potential effects of differential disturbance from weeding these treatments (if indeed they were weeded throughout the experimental period).

The species richness treatments are compared with a separate dataset on natural succession. No information is given about these plots, the soil type, the past management, how many samples there are, how there were sampled, etc. The assertion that rates of C sequestration are higher in the species richness treatments and this is transferable to other sites seems to require a big leap of faith. What special characteristics do planted native species have that would infer higher soil C sequestration rates? Root biomass data look like a scatter, and relationships are quite weak.

It would be helpful to give statistics for differences among treatments, as opposed to just regression analyses, which could be influenced by sample sizes. It is unclear if samples and treatments were confounded.

Specific comments:

Many of the references on soil C dynamics are old outdated.

L13: about half? What is the basis for this gross overgeneralization? Certainly not the reference given, which is referring to a global average (which in itself is not very quantitative).

L30: drop "has"

L31: why compare a cumulative flux over 148 y to an annual flux now?

L38: not all native ecosystems are diverse

L38-40: why would you assume that native species or "native diversity" would increase soil C

storage above non-native species or diversity? Why would this be slower during natural succession? Plant functional traits that lead to soil C storage are not merely a function of diversity or native species.

Fig 1: the results refer to the number of species planted more than 20 years prior, but not to the current species composition or diversity. Are these data available? There is no ecological explanation for the number of species increasing soil C stocks. Information on species traits (allocation to belowground biomass, tissue chemistry, phenology), plant cover and abundance, and soil characteristics (e.g. bulk density, texture, drainage, water holding capacity) would be needed to better understand the patterns observed. It is also apparent that both soil C stocks (1c) and root C stocks (2a) were different across the treatments at the start of the experiment.

L115, L118, L121: what is the error around these mean values? What is the range? What is the effect of species richness and species composition on the values for natural succession plots?

Sup Fig 1: this is not a strong relationship between root C and soil C.

L137-138: and in other cases it has been shown to have no effect.

L143-4: are these statistically significant results? Have you explored sampling effect errors?

L155-157: The statements regarding BECCS seem out of place here. What is the impact of harvesting native restored grasslands for bioenergy on soil C stocks?

Methods: were the plots weeded throughout the 22 to 24-year study? Were the species compositions and diversity levels maintained every year? What is the effect of that level of disturbance on C dynamics? Were all species treatments weeded at the same intensity?

Reviewer #2 (Remarks to the Author):

This manuscript reports rates of soil carbon (C) accrual from a long-term experiment in which levels of plant species diversity were manipulated in a former agricultural field at the Cedar Creek LTER site. The results suggest that rates of soil C accumulation and standing stocks of soil C accumulated over time are a function of plant species richness and associated differences in root biomass C pools under different levels of plant species richness. This is a potentially important and valuable message, and there is considerable interest in using abandoned agricultural lands to both restore native plant diversity and to enhance specific ecosystem services, such as C sequestration. In that regard, it is hard not to support the main conclusions and recommendations of this paper. However, there may be some methodological artifacts of this approach that could limit the broad applicability of conclusions from this specific study. I highlight some of my concerns/questions below, and would like to see these potential limitations addressed in a transparent way if this paper is published, since I think that will be important to readers considering how to apply this information.

My first concern is about the degree to which the pre-planting field treatment and the post-planting removal of non-seeded volunteer species (weeding 3-4 times per season) affected the reported rates of C accumulation, and how that might affect the general applicability of these results to other more typical restoration efforts. The pre-planting treatment included herbicide, burning, removal of the top 6–8 cm of soil, and repeated repeatedly plowing and harrowing. Even in these sandy, low organic matter soils, this would likely have reduced existing C pools to an extreme level. How might this have affected the subsequent measured rates of soil C accrual? Similarly, what affect did the removal of volunteer species have? It may be that greater recruitment of weedy species and other volunteer species early in a more typically managed

restoration would compensate for some of the lower soil C accumulation in the low diversity treatments observed in this experiment. I think this might warrant some consideration and comment in the manuscript. This could be done in a way that doesn't detract from the main message regarding importance of plant diversity, but still acknowledges that soil C accrual rates in real world restorations may not be constrained in the same way.

A second concern is potential confounding effects of the diversity treatments. In experiments such as this, there are always questions regarding whether the observed effects are due to plant diversity per se, or to other factors that may be confounded with diversity (selection effects, species-specific characteristics, etc.). For example, there were likely important correlations between diversity and plant density, particularly in the early establishment phases of the experiment when some of the single species plantings consisted of forbs that generally do not grow densely or produce a lot of above- or below-ground biomass. I know that seeds of all species/species combinations were planted at equal seed biomass per unit area, but that doesn't mean that establishing plant densities were equivalent among diversity treatments. It would be really interesting to see if the single species plantings comprised of the dominant grass species had similar effects on soil C accumulation relative to the subdominant forbs, for example. I would guess that there may have been some species/lifeform differences in root biomass and in soil C accumulation. Can the authors analyze the data that way, or comment on this possibility?

Lines 104-106 – I don't understand the reported significance of the negative relationship of root C stocks with annual precipitation. This is the only reference to precipitation, or other climatic drivers, in the manuscript. Is annual precipitation amount a good predictor of root biomass in a perennial grassland? If so, why? I would expect that a single good/bad rainfall year would not have very direct effects on root biomass C, and I wouldn't expect that relationship to be negative. Why include precipitation here, but not in the analyses of soil C pools or accumulation rates? Could the authors clarify this?

Lines 127-141 – I'm not sure I understand the rationale for why annual soil C accumulation rates should be a fixed proportion (~11%) of measured root C stocks. What is the mechanistic link between root mass of C (a standing stock measured at a point in time) and annual rates of C accumulation (a process that depends on both inputs and losses of C)? I also don't understand the argument that as root standing stocks equilibrate at roughly their current level, soil C stocks will continue to increase at the same rate as in the second decade of the study (lines 134-137). Clearly, both root C stocks and soil C stocks will eventually reach equilibrium levels. At that point, there should be no relationship between root C mass and soil C accumulation rates, right? It may take many decades for annual soil C losses to equal root inputs (comparisons with regional successional fields suggest this), but it is impossible to know whether root C accumulation rates will continue at the present rate, or begin to slow as C levels increase in the next decade. One might conclude that if soil C accrual rates are much higher in high diversity plantings, they might also reach equilibrium sooner. Perhaps the authors could temper or qualify this prediction.

Some additional minor comments follow:

The dashed lines in Figure 1a and b are not necessary and really not that informative.

Line 157 – There does not appear to be a "30" in the listed references. In general, the formatting of the listed references needs to be checked for consistency.

The figures for soil C stocks and rates of accumulation in the text are based on dividing the data into two discrete time periods (first and second decade). However, the tables in the Supplementary Material appear to have analyzed time of sampling based on years of recovery within those decadal intervals "T – number of years since 1993". Can the authors clarify the rationale for this? Also, I will note that the "fit" of these models, although statistically significant, are generally quite low ($R^2 = 0.195$ for soil C at 0-20 cm; $R^2 = 0.107$ for soil C at 0-60 cm; $R^2 =$

0.183 for root C). Given the fairly low explanatory power of these models, perhaps these values should be presented in the main text or figures, in addition to levels of significance.

Responses to reviewers

Reviewer #1

1. This manuscript describes soil C stocks over 22 to 24 years in an old field in MN, USA. The site is part of a well-known research project where researchers attempted to manipulate plant species richness to determine the effects of richness on a suite of ecosystem characteristics. The species richness treatments are repurposed here as a “restoration” experiment. In the present study, soils were sampled four times post treatment and root biomass was sampled three times. The authors argue that the more species-rich plots had more soil C, and thus that enhanced biodiversity during native species restoration leads to greater soil C sequestration.

Reply: Thanks for your comments and overall assessment of our study. The experiment was designed to study the effects of plant diversity and, as you note, can give insights into the restoration of biodiversity.

2a. The manuscript left me with several questions. First, the authors report C stocks, but not bulk density or C concentrations. Neither are mentioned in the manuscript. Soil C stocks can vary as a result of C concentrations, bulk density, or both. Thus, the absence of these data is a critical omission. Bulk density and C concentrations could vary across space, time, and treatments. Rates and not stocks are reported in figure 1, but how the rates were determined is not reported. I could not determine how many samples were collected for each depth at each time point.

Reply: Thanks for your insightful comments, which pointed out the need for direct measurements of bulk density in these plots. We have now measured bulk density in this experiment, and have found, as you suggested, that bulk density is lower in soils with greater soil C. Using these new bulk density measurements, and bulk density measurements taken more than a decade ago, we re-estimated soil C storage rates using the equivalent soil mass approach to account for the observed decreasing bulk density. We now report these new carbon storage results, which are slightly smaller than the previous results, but these changes do not affect our main finding of accelerating soil C storage rates. Please see our revised method section. We provided relevant data in the supplementary tables and added to the text that the raw data of this study can be found at the Cedar Creek Ecosystem Science Reserve website: <https://www.cedarcreek.umn.edu/>

2b. Soils were dried that 40 °C; were dry weights determined at the standard 105 °C? If not, masses could differ as a result of moisture retention in more organic or clay-rich sites, skewing the results.

Reply: Thanks for your comments. Soils were initially dried at 40 °C in a dehumidified “drying room” large enough to accommodate all the soil cores that were collected. However, all reported soil masses, including those for bulk density and for analysis of soil C, were determined after drying in a drying oven at 105 °C. We now make this clear in our revised text.

3a. The authors report species richness of treatments that were established in 1994. However, no

information is given about whether species richness and composition treatments were maintained throughout the whole experiment.

Reply: Thanks for your comments. In the Methods section we now state that “Plots have never been fertilized and have been weeded 3 or 4 times a year to maintain the intended species compositions and diversity.”

3b. No information is given about the density of individuals, or the functional groups or species traits within and across treatments.

Reply: Thanks for your comments. We did not measure the “density” of individual species in terms of the number of individual plants per m². Many of these plant species spread by rhizome (underground stems), making it extremely difficult to determine what constitutes an “individual” plant. Rather, we measured abundances as the dry grams of biomass/m² for each species. As to functional groups, we now include analyses of the effects of four functional groups, in particular the effects of the presence or absence (or biomass) of C4 grass species, of C3 grass species, of legume species, and of non-legume forb species. In response to your comments, we added a table that gives the names and functional group of all of the plant species used in the experiment (Supplementary Table 11).

3c. The authors do not explore the potential for sampling effect (the potential to select a species with traits that heavily influence the results in higher richness treatments). This could easily influence the results for soil C and root biomass.

Reply: Thanks for your comments. We agree that the possible roles of sampling effects and of complementarity merit consideration. We now include a new figure (Figure 3) that allows visual analysis of the roles of sampling versus complementarity, and discuss this issue (lines 173-190).

3d. The authors also do not consider the potential effects of differential disturbance from weeding these treatments (if indeed they were weeded throughout the experimental period).

Reply: Thanks for raising this concern, which we now address. As mentioned before, plots were weeded to maintain experimental species compositions and diversities. Although this experiment did not include a “no-weeding” treatment, another high-diversity experiment in this same field (randomly intermingled with the plots of our experiment) consists of 32-species plots that are not weeded. These 32 species plots stored soil C at rates similar to those of the 16-species plots in our experiment. We now discuss the potential effects of weeding (lines 242-251) and present the results from the 32-species experiment (Supplementary Fig. 2).

4a. The species richness treatments are compared with a separate dataset on natural succession. No information is given about these plots, the soil type, the past management, how many samples there are, how there were sampled, etc.

Reply: Thanks. We have modified our paper by adding an explanation of the successional study that has been underway at our site since 1982. We apologize that we had mis-referenced this study, which is now corrected to be reference 20 (Knops and Bradley 2009). Because the successional study is done at our site, it has similar soils (sandy with soil C concentration <1%).

The grassland species used in our experiment are often the dominant plant species in the late successional fields (> 50 years since abandonment). We now cite the key result of the study (by Knops and Bradley 2009) of soil C during succession as “Analysis of soil C along this chronosequence (based on ~2000 plots from 21 agricultural fields abandoned at different times from 4 to 74 years ago), gives an annual rate of C storage of 0.17 (± 0.05) Mg-C ha⁻¹ yr⁻¹ for the 0-20 cm depth profile and suggests that this annual rate of soil C accumulation under natural succession is approximately constant across this 70 year period²³.”

4b. The assertion that rates of C sequestration are higher in the species richness treatments and this is transferable to other sites seems to require a big leap of faith.

Reply: Thanks for this comment. We did not mean to imply that our results were directly applicable to other sites. We have modified this statement to now read as “The increases in soil C sequestration rates that we observed following restoration of highly diverse plant communities suggest that similar studies should be performed in other sites to determine if or how differences in climate, soil type, plant functional traits may influence soil C storage rates on abandoned lands.”

4c. What special characteristics do planted native species have that would infer higher soil C sequestration rates?

Reply: Our experiment focused on late-successional native species, but we agree with you that our findings may not be limited to such species. Non-native but highly productive C4 grasses like *Miscanthus* also lead to significant soil C sequestration. In response to your comment, we added a sentence in the Discussion to acknowledge the possible role of non-native species (lines 251-254). Moreover, we have modified the paper to emphasize the late-successional status of the planted species rather than their status as native species. As now discussed, the traits of late-successional species seem more likely to contribute to C storage, while being native or exotic may be less relevant.

4d. Root biomass data look like a scatter, and relationships are quite weak.

Reply: We removed this part, and replaced it with analyses of the effects of functional group composition, species composition, and functional traits on soil C storage. Please see Figure 4.

5. It would be helpful to give statistics for differences among treatments, as opposed to just regression analyses, which could be influenced by sample sizes. It is unclear if samples and treatments were confounded.

Reply: Our experiment was explicitly designed as a regression experiment (not an ANOVA experiment) so as to reveal the shape of the dependence of response variables on plant diversity (for example, Fig. 1a, b; Fig. 2b; Fig. 3). By being a long-term experiment, it also reveals temporal dynamics. The power of regression experiments comes from the combination of the number of treatment levels and their degree of replication.

Specific comments:

6. Many of the references on soil C dynamics are old outdated.

Reply: Thanks. We now cite some recent publications (see below), but feel that some of the older references are still relevant.

Wei, Xiaorong, et al. "Global pattern of soil carbon losses due to the conversion of forests to agricultural land." *Scientific reports* 4 (2014): 4062.

Sanderman, Jonathan, Tomislav Hengl, and Gregory J. Fiske. "Soil carbon debt of 12,000 years of human land use." *Proceedings of the National Academy of Sciences* 114.36 (2017): 9575-9580.

7. L13: about half? What is the basis for this gross overgeneralization? Certainly not the reference given, which is referring to a global average (which in itself is not very quantitative).

Reply: Thanks. We cite new references, each showing a range of %C lost from agricultural soils, and now summarized these as 20-67%.

8. L30: drop "has"

Reply: Corrected, thanks.

9. L31: why compare a cumulative flux over 148 y to an annual flux now?

Reply: We have reworded this part and added a more recent estimate in response to your suggestion. We now avoid a direct comparison between the long-term cumulative loss and the annual flux. Our purpose is to show that restoration of even a part of the lost soil C could help "buy time" for the development of low-carbon technologies as suggested by others (e.g., Lal 2004 cited in the text).

10. L38: not all native ecosystems are diverse

11. L38-40: why would you assume that native species or "native diversity" would increase soil C storage above non-native species or diversity? Why would this be slower during natural succession? Plant functional traits that lead to soil C storage are not merely a function of diversity or native species.

Reply: Thanks for these comments. We had not explained our logic well. We have changed how we introduce the hypothesis that we are testing. We now say "Since the original soils of these abandoned lands had been formed by native, late-successional and often highly-diverse ecosystems, we decided to test the possibility that rapid restoration of late successional plant diversity might accelerate soil C storage above the rates observed during natural succession."

We now articulate the logical basis of our hypothesis in the next paragraph: "During the first decades of succession, abandoned agricultural lands are dominated by ... annuals and... C3 plant species. These species are gradually outcompeted by native perennial prairie C4 grass species. Because the C3 grass and forb species have fine roots that decompose 55% and 138% faster, respectively, than do the fine roots of the native C4 grasses, it seems plausible that soil C storage rates would increase when native C4 grasses attain dominance."

12a. Fig 1: the results refer to the number of species planted more than 20 years prior, but not to the current species composition or diversity. Are these data available?

Reply: Our experiment is designed to determine the effects of the diversity of the late-successional grassland species that we planted. The number of planted species is a much better way to measure the diversity of late successional species than a simple tally of the total number of all species – including rare weedy, early successional plants – observed in a plot. We have sampled the vegetation of the plots most years of the experiment since 1996. We find that the effective number of species observed in each plot is highly correlated with the number of planted species ($r^2=0.64$, $P<0.0001$). Moreover, weeding removes non-planted species that invade a plot, meaning that any non-planted species in a plot tend to be rare. When we do analyses comparable to those of Figs 1-4, but using observed species diversity rather than planted diversity, we find results comparable to those we show. However, observed diversity is made “noisier” by counting the presence of rare weedy, early-successional plants.

12b. There is no ecological explanation for the number of species increasing soil C stocks. Information on species traits (allocation to belowground biomass, tissue chemistry, phenology), plant cover and abundance, and soil characteristics (e.g. bulk density, texture, drainage, water holding capacity) would be needed to better understand the patterns observed.

Reply: Thanks for this excellent question. We now present information on plant functional groups, root:shoot ratios, root mass and root decomposition rates, and do new analyses to explore the potential mechanistic basis of the higher C storage rates. Please see Figure 4 and related text.

12c. It is also apparent that both soil C stocks (1c) and root C stocks (2a) were different across the treatments at the start of the experiment.

Reply: Prior to planting, soil C concentration (Fig. 1c, which shows pre-planting soil C levels) differed slightly (from 0.46% to 0.52%), but not significantly, among the randomly-located treatment replicate plots. Please note that C storage rates are calculated using the difference between the pre-planting soil C levels and the soil C levels measured various times after planting. Because of the field was plowed and repeatedly disked the year prior to planting, root biomass was zero in all plots prior to planting. When we first measured root mass, in the third growing season, 1996 (Fig. 2a), plots planted with higher plant diversity had achieved greater root mass than those with lower diversity.

13. L115, L118, L121: what is the error around these mean values? What is the range? What is the effect of species richness and species composition on the values for natural succession plots?

Reply: Good questions. Thanks. We’ve now added standard errors for all means to the graphs and text. They are also shown in the supplementary tables we now include. We also calculated standard errors for the result from the natural succession experiment carried out at our site (Knops and Bradley 2009). We now say (lines 149-153) that: “Analysis of soil C along this chronosequence (based on ~2000 plots from 21 agricultural fields abandoned at different times from 4 to 74 years ago), gives an annual rate of C storage of $0.17 (\pm 0.05) \text{ Mg-C ha}^{-1} \text{ yr}^{-1}$ for the

0-20 cm depth profile and suggests that this annual rate of soil C accumulation under natural succession is approximately constant across this 70 year period.”

Their paper did not explore the possible effects of plant diversity or composition on soil C storage rates during succession.

14. Sup Fig 1: this is not a strong relationship between root C and soil C.

Reply: Thanks for the comments. We removed this simple univariate analysis, and replaced it with more insightful analyses that include functional composition and traits. Please see Figure 4 and related text.

15. L137-138: and in other cases it has been shown to have no effect.

Reply: We have removed this sentence.

16. L143-4: are these statistically significant results? Have you explored sampling effect errors?

Reply: Supplementary Table 4 gives, for each diversity level, means and standard errors for the annual percent C storage rates. For the 16-species diversity treatment, the rate of increase is 1.6% yr⁻¹ with a standard error of 0.2. Based on comments by a different reviewer, we no longer compare this rate with the 0.4% “goal.”

17. L155-157: The statements regarding BECCS seem out of place here. What is the impact of harvesting native restored grasslands for bioenergy on soil C stocks?

Reply: We agree, and have deleted discussion BECCS and added more relevant references for bioenergy and ecological restoration (Awasthi et al. 2017; Nackley et al. 2013). Harvesting seems to have little effect on soil C; please see, for example, Gelfand et al. 2013.

18. Methods: were the plots weeded throughout the 22 to 24-year study? Were the species compositions and diversity levels maintained every year? What is the effect of that level of disturbance on C dynamics? Were all species treatments weeded at the same intensity?

Reply: Thanks. We now describe that all plots were annually weeded several times per year, and discuss the potential impacts of this weeding on lines 242-251.

Reviewer #2

1. This manuscript reports rates of soil carbon (C) accrual from a long-term experiment in which levels of plant species diversity were manipulated in a former agricultural field at the Cedar Creek LTER site. The results suggest that rates of soil C accumulation and standing stocks of soil C accumulated over time are a function of plant species richness and associated differences in root biomass C pools under different levels of plant species richness. This is a potentially important and valuable message, and there is considerable interest in using abandoned agricultural lands to both restore native plant diversity and to enhance specific ecosystem services, such as C sequestration. In that regard, it is hard not to support the main conclusions

and recommendations of this paper. However, there may be some methodological artifacts of this approach that could limit the broad applicability of conclusions from this specific study. I highlight some of my concerns/questions below, and would like to see these potential limitations addressed in a transparent way if this paper is published, since I think that will be important to readers considering how to apply this information.

Reply: Thanks for your overall positive assessment of our study and for your suggestions. We have added a penultimate paragraph that discusses these concerns (lines 237-254).

2a. My first concern is about the degree to which the pre-planting field treatment and the post-planting removal of non-seeded volunteer species (weeding 3-4 times per season) affected the reported rates of C accumulation, and how that might affect the general applicability of these results to other more typical restoration efforts. The pre-planting treatment included herbicide, burning, removal of the top 6–8 cm of soil, and repeated repeatedly plowing and harrowing. Even in these sandy, low organic matter soils, this would likely have reduced existing C pools to an extreme level. How might this have affected the subsequent measured rates of soil C accrual?

Reply: Thanks for your comments. We agree that the initial soil C concentration of our plots was low, at around 0.5% (Supplementary Table 1). This low initial C level may contribute to the high annual growth rate of soil C that we observed (now calculated using $dC/dt * 1/C$) in the high-diversity treatment. For this reason we no longer emphasize this rate, and deleted the sentence where it was mentioned.

2b. Similarly, what affect did the removal of volunteer species have? It may be that greater recruitment of weedy species and other volunteer species early in a more typically managed restoration would compensate for some of the lower soil C accumulation in the low diversity treatments observed in this experiment. I think this might warrant some consideration and comment in the manuscript. This could be done in a way that doesn't detract from the main message regarding importance of plant diversity, but still acknowledges that soil C accrual rates in real world restorations may not be constrained in the same way.

Reply: Thanks for your comments on the possible effects of weeding on soil C storage. We now discuss this issue in more detail, saying in the penultimate paragraph that: “As in other experiments^{30,31}, weeds appeared much more frequently and in higher density in our lower-diversity plots. Weeding removes biomass, and thus might lower soil C storage rates. However, if competition with weeds reduces the productivity of the planted late-successional species³², weeding might increase soil C storage because of greater abundance of late-successional species. In a parallel experiment at our site, plots that had been planted with 32 late-successional grassland species and never weeded had remarkably similar rates of soil C sequestration as our 16-species plots (Supplementary Fig. 2), suggesting that high diversity of late-successional species leads to high rates of soil C storage even without weeding.”

30. Abernathy, J. E., Graham, D. R. J., Sherrard, M. E. & Smith, D. D. Productivity and resistance to weed invasion in four prairie biomass feedstocks with different diversity. *GCB Bioenergy* **8**, 1082–1092 (2016).

31. Blumenthal, D. M., Jordan, N. R. & Svenson, E. L. Weed Control as a Rationale for Restoration: The Example of Tallgrass Prairie. *Conserv. Ecol.* **7**, 6 (2003).

3. A second concern is potential confounding effects of the diversity treatments. In experiments such as this, there are always questions regarding whether the observed effects are due to plant diversity per se, or to other factors that may be confounded with diversity (selection effects, species-specific characteristics, etc.). For example, there were likely important correlations between diversity and plant density, particularly in the early establishment phases of the experiment when some of the single species plantings consisted of forbs that generally do not grow densely or produce a lot of above- or below-ground biomass. I know that seeds of all species/species combinations were planted at equal seed biomass per unit area, but that doesn't mean that establishing plant densities were equivalent among diversity treatments. It would be really interesting to see if the single species plantings comprised of the dominant grass species had similar effects on soil C accumulation relative to the subdominant forbs, for example. I would guess that there may have been some species/lifeform differences in root biomass and in soil C accumulation. Can the authors analyze the data that way, or comment on this possibility?

Reply: Thanks for these excellent comments and suggestions. We have done additional analysis, and added two new figures (Figs. 3 & 4) that address these issues. As to selection (sampling) effects, Figure 3b shows that the species with the greatest root mass in monoculture had less root mass than every one of the 16 species plots. This result shows that selection effects are small relative to effects that result from multi-species interactions. Figure 4 shows how C3 grasses, C4 grasses, legumes and forbs differ in their root masses and in root decomposition rates, and how C4 grasses and legumes impact soil C storage rates.

4. Lines 104-106 – I don't understand the reported significance of the negative relationship of root C stocks with annual precipitation. This is the only reference to precipitation, or other climatic drivers, in the manuscript. Is annual precipitation amount a good predictor of root biomass in a perennial grassland? If so, why? I would expect that a single good/bad rainfall year would not have very direct effects on root biomass C, and I wouldn't expect that relationship to be negative. Why include precipitation here, but not in the analyses of soil C pools or accumulation rates? Could the authors clarify this?

Reply: Thanks for raising these questions. Because we did not detect a significant effect of precipitation in any of our other analyses, we agree that precipitation does not add much, if any, insight. We removed the variable from this regression and from our paper.

5. Lines 127-141 – I'm not sure I understand the rationale for why annual soil C accumulation rates should be a fixed proportion (~11%) of measured root C stocks. What is the mechanistic link between root mass of C (a standing stock measured at a point in time) and annual rates of C accumulation (a process that depends on both inputs and losses of C)? I also don't understand the argument that as root standing stocks equilibrate at roughly their current level, soil C stocks will continue to increase at the same rate as in the second decade of the study (lines 134-137). Clearly, both root C stocks and soil C stocks will eventually reach equilibrium levels. At that point, there should be no relationship between root C mass and soil C accumulation rates, right? It may take many decades for annual soil C losses to equal root inputs (comparisons with regional successional fields suggest this), but it is impossible to know whether root C accumulation rates will continue at the present rate, or begin to slow as C levels increase in the next decade. One

might conclude that if soil C accrual rates are much higher in high diversity plantings, they might also reach equilibrium sooner. Perhaps the authors could temper or qualify this prediction.

Reply: Again, thanks for these interesting questions and comments. We had found it interesting that the annual accrual rate of soil C was ~10% of the C content of roots. Since root:shoot ratios in these grasslands are ~5.5, roots seemed, to us, to be a likely source for most of the C accumulating in the soil. However, we agree with you that our extrapolations were poorly justified and tangential to our main points. We have thus removed these speculations from our paper, and replaced them with what we feel are more rigorous and insightful analyses (such as Figures 3 and 4).

Some additional minor comments follow:

6. The dashed lines in Figure 1a and b are not necessary and really not that informative.

Reply: Thanks. We removed the dashed lines.

7. Line 157 – There does not appear to be a “30” in the listed references. In general, the formatting of the listed references needs to be checked for consistency.

Reply: Thanks. We have corrected the reference list.

8a. The figures for soil C stocks and rates of accumulation in the text are based on dividing the data into two discrete time periods (first and second decade). However, the tables in the Supplementary Material appear to have analyzed time of sampling based on years of recovery within those decadal intervals “T – number of years since 1993”. Can the authors clarify the rationale for this?

Reply: Thanks for finding our labeling error in the former Supplementary Table 2 (which is now Supplementary Table 6). The “time” variable for that analysis should have been listed as the categorical variable “decade”. This is now corrected.

To compare C storage rates for the first and second “decades”, we analyzed rates for each of two time periods (1994-2006 and 2006-2015). In contrast, to determine the temporal dynamics of C storage, we analyzed soil C concentrations for every year in which they were measured, and included year as variable. Both ways of analysis support our finding that rates of soil C storage increased during this 22 year-long study.

Also, I will note that the “fit” of these models, although statistically significant, are generally quite low ($R^2 = 0.195$ for soil C at 0-20 cm; $R^2 = 0.107$ for soil C at 0-60 cm; $R^2 = 0.183$ for root C). Given the fairly low explanatory power of these models, perhaps these values should be presented in the main text or figures, in addition to levels of significance.

Reply: Thanks for the suggestion. We now included these r^2 values in the text, and present the full statistical analysis in the supporting tables.

REVIEWERS' COMMENTS:

Reviewer #1 (Remarks to the Author):

The paper is much improved with the inclusion of basic soil parameters such as bulk density and a clarification of the methods. I have only a few remaining issues listed below.

L 57-58: rates of decomposition do not necessarily affect soil C stocks. Root turnover is much more important here and it appears you have the data to provide this. Decomposition is usually reported as k values, not percent. What does the percent refer to?

L74-74: year 13 is the second decade and year 22 is the third decade. Just give the years or refer to them as periods to be accurate throughout the manuscript; how did you chose 1-13 (13 years) versus 13-22 (9 years) as the periods of examination?

L 207-209: you state that soil C increases with plant biomass, but only report the P value in the text (for a very large sample size that increases the likelihood of a significant result). However, looking in the supplement, the coefficient is very low. Thus, it does not appear that biomass is a strong predictor of soil C concentrations.

L 231: add pools after C

L 423: what years were root sampled? Were the roots separated by live and dead categories?

L 429: Costech is the company, not the analytical instrument. Please give the instrument type.

Reviewer #2 (Remarks to the Author):

This manuscript is a revised version of one that I reviewed previously. As I noted before, there is considerable interest in using abandoned agricultural lands to both restore native plant diversity and to enhance specific ecosystem services, such as C sequestration, and the topic of how species richness in revegetated abandoned agricultural lands affects the magnitude and timing of soil C accumulation is both timely and important. However, both I and another reviewer raised several questions regarding some of the field and analytical methods that could have influenced the results of the study and the general applicability of the conclusions from this specific study to other restoration efforts. I pleased to note that the authors adequately addressed the concerns and questions I raised in my prior review. As a result, I find this manuscript greatly improved. In particular, I appreciate the greater clarity and transparency regarding methodological details, including pre-planting field treatment and the effects of post-planting removal of non-seeded volunteer species, and how this may have altered reported rates of C accumulation. I find their rationale for concluding that this was not a major factor satisfactory, particularly with the addition of information from the non-weeded high-diversity plots in the same general area. I also appreciate the improved descriptions of the successional field data they used for comparative purposes.

I also raised concerns in my prior review about the potential impact of selection effects, as did another reviewer. In my opinion, they did a good job of acknowledging this concern, and also providing data to suggest that the greater rates of C accumulation were more likely due to complementarity (Fig. 3 and the added text on biodiversity and sampling vs. complementarity). In my opinion, this helps strengthen their conclusions. I also appreciate the new analyses of how plant life-form and associated traits (relative root biomass, root decomposability) may have contributed to the differences in rates among high- and low-diversity treatments. The correlation of C accrual rates with specific species (notably three C4 grasses and two legumes) is also a nice

addition.

In general, I found the revised manuscript to be well-written and much clearer in terms of methodological details. I also think that the authors added appropriate caveats about the need to test the general conclusions in other ecosystems, soils, etc. I have two minor, specific editorial suggestions for clarity, but otherwise I think this is acceptable as is.

Line 124 –I suggest adding “aboveground” before productivity in this sentence to clarify that ANPP varied year-to-year with climate drivers.

Line 231 – add “accrual” (...caused soil C “accrual” to accelerate...)

Responses to reviewers

(with quoted reviewer text, followed by our response as “Reply: ...”)

Referee report, Reviewer 1:

Reviewer #1 (Remarks to the Author):

The paper is much improved with the inclusion of basic soil parameters such as bulk density and a clarification of the methods. I have only a few remaining issues listed below.

Reply: Thanks for the positive assessment of our revision.

L 57-58: rates of decomposition do not necessarily affect soil C stocks. Root turnover is much more important here and it appears you have the data to provide this. Decomposition is usually reported as k values, not percent. What does the percent refer to?

Reply: Thanks for the comments. We do not have root turnover data, but do have the percent of fine root mass lost via decomposition during a field incubation. Because of the diverse audience that might read our paper, we feel that the percent lost to decomposition will be more informative than k values. We now make this clear in the caption, stating “Fine root decomposition percentage of different functional groups (measured after 10 months of field incubation, which included ~5 winter months²⁹).” This field decomposition study was sampled only once, at the end of a 10-month period, of which ~ 5 months were in the winter when soils were frozen at our northern site.

L74-74: year 13 is the second decade and year 22 is the third decade. Just give the years or refer to them as periods to be accurate throughout the manuscript; how did you chose 1-13 (13 years) versus 13-22 (9 years) as the periods of examination?

Reply: Thanks for the suggestion. We have revised the full paper accordingly. The choice of 1-13 years (13 years) versus 13-22 (9 years) was because soil carbon for the entire 60 cm of soil was only sampled before the planting (spring of 1994), in year 13 (fall of 2006), and in year 22 (fall of 2015).

L 207-209: you state that soil C increases with plant biomass, but only report the P value in the text (for a very large sample size that increases the likelihood of a significant result). However, looking in the supplement, the coefficient is very low. Thus, it does not appear that biomass is a strong predictor of soil C concentrations.

Reply: As noted in various places in our paper, soil C storage is statistically associated with multiple variables, including the joint presence of C4 grasses and legumes, root mass, time and diversity. We did not mean to imply that plant biomass was the major determinant, but rather to note that it was a correlate of greater C storage rates. We have modified this text to now read “Soil C sequestration was positively associated with aboveground plant biomass and root biomass. A linear mixed model showed that soil C concentrations (as %) in the upper 20 cm of soil were positively correlated with root biomass (P=0.0004), time (P<0.0001) and aboveground plant biomass (P=0.0033; Supplementary Table 10).”

L 231: add pools after C

Reply: Corrected as “...with C pools accumulating...”, thanks.

L 423: what years were root sampled? Were the roots separated by live and dead categories?

Reply: Thanks for the questions. The roots were sampled in 12 different years, as indicated by dots in figure 2a. We now reference Figure 2a in this sentence. We did not explicitly sort dead roots from live, but our root washing process tends to flush out dead roots because of how fragile they are.

L 429: Costech is the company, not the analytical instrument. Please give the instrument type.

Reply: As now noted in the revised paper, analyses were done using a Costech ECS 4010 Analyzer.

Referee report, Reviewer #2:

This manuscript is a revised version of one that I reviewed previously. As I noted before, there is considerable interest in using abandoned agricultural lands to both restore native plant diversity and to enhance specific ecosystem services, such as C sequestration, and the topic of how species richness in revegetated abandoned agricultural lands affects the magnitude and timing of soil C accumulation is both timely and important. However, both I and another reviewer raised several questions regarding some of the field and analytical methods that could have influenced the results of the study and the general applicability of the conclusions from this specific study to other restoration efforts. I pleased to note that the authors adequately addressed the concerns and questions I raised in my prior review. As a result, I find this manuscript greatly improved. In particular, I appreciate the greater clarity and transparency regarding methodological details, including pre-planting field treatment and the effects of post-planting removal of non-seeded volunteer species, and how this may have altered reported rates of C accumulation. I find their rationale for concluding that this was not a major factor satisfactory, particularly with the addition of information from the non-weeded high-diversity plots in the same general area. I also appreciate the improved descriptions of the successional field data they used for comparative purposes.

I also raised concerns in my prior review about the potential impact of selection effects, as did another reviewer. In my opinion, they did a good job of acknowledging this concern, and also providing data to suggest that the greater rates of C accumulation were more likely due to complementarity (Fig. 3 and the added text on biodiversity and sampling vs. complementarity). In my opinion, this helps strengthen their conclusions. I also appreciate the new analyses of how plant life-form and associated traits (relative root biomass, root decomposability) may have contributed to the differences in rates among high- and low-diversity treatments. The correlation of C accrual rates with specific species (notably three C4 grasses and two legumes) is also a nice addition.

In general, I found the revised manuscript to be well-written and much clearer in terms of methodological details. I also think that the authors added appropriate caveats about the need to test the general conclusions in other ecosystems, soils, etc. I have two minor, specific editorial suggestions for clarity, but otherwise I think this is acceptable as is.

Reply: Thanks for your positive assessment.

Line 124 –I suggest adding “aboveground” before productivity in this sentence to clarify that ANPP varied year-to-year with climate drivers.

Reply: Added as suggested, thanks.

Line 231 – add “accrual” (...caused soil C “accrual” to accelerate...)

Reply: Added as suggested, thanks.